# Resource Threat versus Resource Loss and Emotional Well-Being of Ethnic Minorities during the COVID-19 Pandemic

**DOI:** 10.3390/ijerph182312590

**Published:** 2021-11-29

**Authors:** Rafael Youngmann, Nonna Kushnirovich

**Affiliations:** 1Ruppin Academic Center, Clinical Psychology Graduate Program, Emek-Hefer 4025000, Israel; 2Ruppin Academic Center, Department of Economics and Management, Emek-Hefer 4025000, Israel; nonna@ruppin.ac.il

**Keywords:** COVID-19, emotional well-being, threats, losses, minorities

## Abstract

This paper used Hobfoll’s conservation of resources theory as a theoretical framework to investigate which kinds of resource loss predicted the emotional well-being (EWB) of ethnic minorities and majority populations during a period of crisis. Data were collected from a national representative survey conducted by the Israel Central Bureau of Statistics during the COVID-19 pandemic. The sample included 1157 respondents, including 174 Israeli Palestinian citizens (ethnic minority) and 983 Israeli Jews (majority population). Measures of EWB, actual losses and threats of losses of economic, social, and health resources were examined. The results showed that the losses of economic, social, and health resources reduced the EWB of individuals. Negative effects of the actual losses of resources on EWB were greater than those of the perceived threats of loss. The largest effect was for economic resources. There were differences in effects between the ethnic minorities and the majority populations. The study revealed that for the ethnic minorities, who are less powerful and more disadvantaged than ethnic majorities, the depletion of already deficient resources during time of crisis is more important for predicting their EWB than for the majority populations.

## 1. Introduction

During crises, people experience losses of economic, social, and health resources as well as threats to these resources [1,2]. The ongoing COVID-19 pandemic is an example of such a crisis, when losses of all kinds of resources and threats to them are widespread.

Reports on negative impacts of the crises on people’s emotional well-being (EWB) are accumulating [3]. EWB is defined as the emotional quality of an individual’s everyday experience in terms of frequencies and intensities of experiences of joy, stress, sadness, anger, and affection [4]. Studies of the psychological status of populations in different countries during the COVID-19 pandemic revealed relatively high levels of depression, anxiety, stress, and psychological distress [5]. The crisis caused by the coronavirus pandemic is multidimensional, where losses and threats to health are accompanied by losses and threats to individuals’ economic and social resources. Anxiety and depressive symptoms are reactions to uncertainty, perceived threats, and multiple losses of resources associated with the pandemic [6].

Minorities usually report lower EWB than majority populations [7,8,9]. The consensus in the literature is that minorities have poorer economic, social, and health resources than majority populations [7,10,11]. Concerning coronavirus morbidity, in England for instance, ethnic minorities are overrepresented among COVID-19 victims [12]. In the United States, COVID-19 death rates among Afro-American and indigenous Americans are twice or more that of Caucasian Americans [13]. The chronic lack of resources makes people more vulnerable to additional losses [14], since they lack resources to offset previous losses, creating so-called “loss spirals” [15]. Accordingly, the effects of the loss of these resources may be more pronounced for disadvantaged minorities.

Numerous studies devoted to the impact of the coronavirus pandemic on EWB in different countries have been published during the last year [2,3,5]. However, most of them confirm a negative pandemic impact on the EWB of the whole population without distinguishing between majority and minority groups, and studies of minorities focus mostly on the incidence of cases and death rates among them. Several studies considering the EWB of minorities versus majority populations during the crisis found that people identified with ethnic minority groups, which even before the pandemic crisis experienced mental health inequalities, have disproportionately higher risks of being emotionally harmed by COVID-19 [16,17]. However, the questions of whether this negative impact stems mainly from actual losses or just from the threat posed by the pandemic and the loss of which type of resources is salient for predicting EWB decrease have not been researched sufficiently. This study aimed to fill that gap. The purpose of this study was to investigate how the actual losses and threats of loss of different resources in times of crisis relate to the EWB of an ethnic minority versus the majority population and what kind of effect is salient for each group. The study is based on the data collected during the coronavirus outbreak, as an example of worldwide crises during which the losses of resources and threats to them are extreme and widespread.

### 1.1. Theoretical Framework

From a psychological viewpoint, resources are defined as “those objects, personal characteristics, conditions, or energies that are valued by the individual or that serve as a means for attainment of these objects, personal characteristics, conditions, or energies” (p. 516) [18]. From a sociological perspective, resources may be described in terms of capital accumulated by an individual. Bourdieu [19] defined three basic forms of capital: economic capital, which may be convertible into money and sometimes is institutionalized in the form of property rights, social capital made up of social connections, and cultural capital in terms of cultural goods and long-lasting dispositions. Strong evidence indicates that health may be added to Bourdieu’s forms of capital [20,21], where health capital is conceptualized as “the aggregate of the actual or potential resources possessed by a given agent” (p. 1) [21]. Like other forms of capital, health may be used as an asset for obtaining returns through the production of healthy time [22]. The loss of economic, social, or health resources reduces the optimism and well-being of individuals, since they are left with fewer resources to cope with new challenges [23].

The coronavirus pandemic is characterized by either actual loss of resources or threats of such a loss. In the spring of 2020, the COVID-19 pandemic resulted in high levels of unemployment [1], higher food prices, and reduced business sales [24]. In the United States, the unemployment rates in April and March 2020 were 14.7 percent and 4.4 percent, respectively; Feeding America estimated a 5 percent increase in the food insecurity rate in April 2020 [24]. All these reflect the loss of economic resources.

Concerning the loss of social resources, quarantine, and lockdowns—the restriction of movement to prevent infection—were found to have a negative effect on the EWB of populations across 10 countries, including manifestations of anger, confusion, and post-traumatic stress symptoms. Stressors included quarantine, infection fears, annoyance, inadequate supplies, deficient information, and stigma [2,25]. The social distancing imposed on the public to prevent the spread of the virus causes loneliness and social isolation which worsen the burden of stress, often with deleterious effects on mental and physical health [2].

Regarding the loss of health resources, a study of all causes of mortality in 21 industrial countries from mid-February to May 2020, the first wave of the pandemic, reveals 206,000 more deaths than would have occurred without the pandemic. Similar numbers of excess deaths, excess deaths per 100,000 people, and relative increase in deaths were found in most countries [26]. In addition, some individuals who survive from the COVID-19 are likely to have significant long-term complications, including respiratory, cardiac, and mental health disorders, and may have an increased risk of premature death [1].

Individuals who did not lose their resources during this pandemic experienced fear and threat of losses. Bad feelings following a threat are often contagious, and fear-incited threats may appear to loom more imminently [27]. The findings of a meta-analysis showed that individuals are more susceptible to strong fear appeals and respond to them more severely than to low or weak levels of fear appeals [28]. A study conducted in eight high-income countries revealed that from 29 May to 12 June 2020, the threats perceived for oneself/family, country/world, and financial loss and mistrust in authorities were associated with reduced EWB, as measured by the incidence of anxiety and depression among adults [29].

The conservation of resource (COR) theory [15] can form a basis for comprehending the processes involved in experiencing, dealing with, and defeating chronic and traumatic stress such as that engendered by the COVID-19 pandemic. According to the COR model, psychological well-being is strongly influenced by the perceived threat of resource loss, the actual resource loss, and the lack of gain following resource investment. Both perceived resource threat and actual resource loss are important factors, the reactions to which are psychological stress and lower EWB [18].

Based on the COR theory, we can hypothesize that:

**Hypothesis** **1** **(H1).**
*There will be negative associations between EWB and health threat (H1.1), economic threat (H1.2), and social threat (H1.3).*


**Hypothesis** **2** **(H2).**
*There will be negative associations between EWB and health loss (H2.1), economic loss (H2.2), and social loss (H2.3).*


Economic and physical threats are usually conceptualized in the literature as realistic, while threats to a group’s social and cultural patterns are conceptualized as symbolic [30]. Due to the pandemic, realistic threats of resource loss manifest themselves in grief, a common human response to loss, which appears to be a primary outcome of COVID-19 [6]. In periods such as the coronavirus crisis, loss can lead to “complicated grief”, in which serious symptoms of grief, to the point of dysfunction, may last for months or years. Threats can lead either to unresolved grief or “ambiguous loss”, a reaction to situations where the unknown seems to exceed the known, or to “anticipatory grief”, a reaction to situations where people also grieve potential future losses. The effects of threats and losses on EWB also may differ. The uncertainty distress model posits that uncertainty is a main cause of distress, even in low-threat situations [31]. Thus, its effect may differ from that of loss, which brings “complicated grief” [6]. On the contrary, Weir [32] claims that whether the loss is actual or anticipated, the reactions are genuine and valid.

A paradoxical situation emerged during the pandemic. Measures intended to mitigate loss of health such as social distancing, lockdowns, closure of shops and restaurants, and cancellation of cultural events actually caused economic and social losses, creating a kind of trade-off effect between them [33]. Foa et al. [34] found that, in Great Britain, the negative mental health effects during the coronavirus crisis were principally the result of the pandemic (loss of health resources) and were not the result of the lockdowns (loss of social resources). Thus, there is a reason to hypothesize that the effects on EWB of losses of and threats to different kinds of resources would vary significantly.

Ethnic minorities have lower economic, social, and health resources than majority populations [7,8]; hence, the loss of resources may be more important for them and their EWB. In economic theories, the term “utility” is used to represent satisfaction, happiness, or EWB. Marginal utility is a term defining the change in utility as an additional unit of a product or resource consumed. According to the law of diminishing marginal utility, a basic concept of economic theory, marginal utility diminishes, when we use more resources, that is, the marginal utility derived from a larger-sized unit of a resource is smaller than the marginal utility derived from a smaller-sized unit of a resource [35]. The law also works in the opposite direction: the loss of one unit of a resource by a person with many such units decreases the utility (EWB) less than the loss of one unit of resource by a person with only a small number of such units. Thus, the effects of loss of resources on EWB may be more important to their EWB.

Social, economic, and health resources are extremely important for minorities. Social support within groups that share ethnicity and cultural backgrounds provides an important resource for minorities [36]. Income may significantly increase EWB and may even moderate the negative effects of discrimination and social exclusion [11]. Some studies attribute minorities’ lower mental health during the pandemic, compared to that of majority populations, to their high rates of mortality and incidence [36], intersected discrimination [37], or poorer mental health before the pandemic [16]. Thus, the effects of threats and losses of different resources may vary across minority and majority groups, as well as across kinds of resources. Based on this, we can formulate the following research questions:

Q1. Which effects on EWB are higher: the effects of resource threats or the effects of resource losses?

Q2. Do the effects of resource threats and losses on EWB differ for minority and majority groups, and which effect is higher for each group?

### 1.2. The Israeli Context

In Israel (population: 9.1 million) [38], the virus was confirmed on 21 February 2020. Within a month or so, the total number of confirmed cases increased to 1000. Very shortly afterwards, confirmed cases exceeded 4000. The Israeli government then implemented extreme lockdown measures which sharply decreased daily infection rates from a peak of 1131 to slightly more than 100 new confirmed cases on 30 April 2020. Encouraged by this improvement, similar data from many other countries, and heightening economic pressures, the Israeli government lifted most of its emergency guidelines almost all at once [39].

Israeli Palestinians, an ethnic minority in the country, constitute about 20 percent of Israel’s population. The differences between Israeli Palestinians and Israel’s Jewish majority manifest in terms of ethnicity, religion, culture, language, and resources [40]. Israeli Palestinians are disadvantaged in terms of income [11] and are socially limited in terms of residence, land ownership, labor market participation, and housing [41]. From February to early June 2020, when the first wave of the pandemic waned, Jewish cities recorded consistently higher morbidity rates than Arab communities. Several factors were cited for the lower Israeli Palestinian morbidity, including spatial Jewish–Palestinian segregation [42]. However, during the second and third waves, morbidity rates for Israeli Palestinians were higher because of crowded social events such as wedding celebrations where social distancing was not enforced [43].

## 2. Materials and Methods

### 2.1. Data

The survey was carried out in May 2020, when the first wave of the pandemic seemed to recede, and the full lockdown and other limitations on the population were lifted. At that time, the Israel Central Bureau of Statistics (CBS) [38] conducted a flash survey, to provide decision-makers with an overview and vital data regarding civil resilience in Israel, following the coronavirus crisis. The survey population included those aged 31 and over, except for the Bedouin diaspora in the south and residents of therapeutic institutions.

The survey sample of 3371 persons was representative of the Israeli population taken from the Israeli Population Registry; 53 percent of the total sample responded to the survey. The final sample of the study included 1157 respondents, including 983 Israeli Jews and 174 Israeli Palestinian citizens. The sex distributions of both groups were rather equal. Israeli Palestinians were younger; 54.0 percent of Israeli Palestinians and 42.5 percent of Israeli Jews were aged 21–44. Israeli Jews were more educated; 33.6 percent of them and 12.1 percent of Israeli Palestinian respondents had academic education. Among Israeli Palestinians, a higher percent of respondents, compared to that of Israeli Jews, was not employed (46.8 percent vs. 35.9 percent).

### 2.2. Measures

EWB was measured using the Center for Epidemiologic Studies Depression Scale (SEC-D) [44] and the construct of subjective well-being developed by Diener et al. [45]. Both included feelings of depression, stress, and anxiety, and the SEC-D also included feeling lonely. During the COVID-19 crisis, feeling lonely is an important component of EWB because of the social distance requirement [25]. In this study, the measure of EWB was based on three items describing feelings of stress and anxiety, being depressed, and feelings of loneliness. The scale ranged from 1 = “to a great extent” to 4 = “not at all”, where higher values indicated higher EWB. The internal reliability value was 0.799. We tested the measure of EWB using confirmatory factor analysis (CFA), treating EWB as a latent variable (see Section 3).

Resource threats during the COVID-19 crisis included four different kinds: the health threat of contracting COVID-19; the health threat of worsening health for other reasons (not COVID-19); the economic threat of difficulties in meeting economic expenses; and the social threat of worsening family relationships. The threats were formulated as follows: “In this period, to what extent do you feel threatened by the following scenarios?” on scales of 1 to 4, where 1 = “not at all” and 4 = “to a large extent”. Higher values indicated higher feelings of threat.

Resource losses due to the COVID-19 crisis were of three kinds. Health loss was formulated as “Compared to your health before the COVID-19 crisis, how has your health changed today?” on scales of 1 to 5, where 1 = “substantially improved” and 5 = “substantially worsened”. Economic loss was formulated as: “Due to the COVID-19 crisis, how has your household’s economic situation changed?” on scales of 1 to 5, where 1 = “substantially improved” and 5 = “substantially worsened”. Social loss was formulated as: “Prolonged stay of family members in the home can create tensions between them. To what extent are there tensions between your household members since the COVID-19 crisis began (including manifestations of violence)?” on scales of 1 to 5, where 1 = “not at all” and 5 = “to a very large extent”. For all these variables, higher values indicated higher loss.

Control variables included sex (coded 1 = male and 0 = female); age represented by two dummy variables (dummy 1 for ages 45–64, dummy 2 for ages 65+, and reference group for ages 21–44); education (coded 1 = “academic” (having the highest diploma of Bachelor’s or higher degree), 0 = “non-academic (the highest diploma less than Bachelor’s degree; this category also included vocational non-academic studies); married (coded 1 = “married” and 0 = “not married”); health status (on scales of 1 to 4, where 1 = “not good at all” and 4 = “very good”); and employment status presented by three dummy variables (dummy 1 for salaried workers, dummy 2 for self-employed, and dummy 3 for workers on furlough, where the reference group was non-employed). We also controlled for trust in government, formulated as: “To what extent do you trust the government to deal with the COVID-19 crisis?” on scales of 1 to 4, where 1 = “not at all” and 4 = “to a large extent”.

Variable definitions and descriptions are presented in Table 1.

## 3. Results

### 3.1. Overview

The factor structure and measurement invariance of EWB were justified by means of CFA. The fit indices of the configural, metric, and scalar models were found to be acceptable (χ^2^/df < 1.575; root-mean-square error of approximation RMSEA < 0.023; comparative fit index CFI > 0.998; Bentler–Bonett normed fit index NFI > 0.992). All item weights were higher than 0.668. The differences in chi-square values between the models were found to be non-significant (Δχ^2^ = 0.887, *p* = 0.642 [2] between the baseline and metric models; and Δχ^2^ = 6.983, *p* = 0.072 [3] between the metric and scalar models). According to Davidov et al. [46], three invariance levels—configural, metric, and scalar—are evidence of measurement invariance. Thus, the constructs of EWB can be legitimately compared across the groups.

Compared to their Jewish counterparts, Israeli Palestinians had lower education (12.1 percent vs. 33.6 percent) and lower rates of employment (42.7 percent vs. 55.4 percent). Thus, Israeli Palestinians may be regarded as a disadvantaged minority lacking resources. The EWB of Israeli Palestinians was lower than the EWB of Israeli Jews (3.23 percent and 3.36 percent, respectively). Israeli Palestinians’ actual economic losses were higher than those of the Jewish population (3.63 percent vs. 3.45 percent), but their social and health losses were similar to those of the majority population (1.65 percent vs. 1.62 percent for social losses; 2.98 percent vs. 2.97 percent for health losses). Israeli Palestinians reported higher levels of economic threat than did Israeli Jews (2.50 percent vs. 2.20 percent), social threat (1.47 percent vs. 1.30 percent), threat of contracting COVID-19 (2.54 percent vs. 2.42 percent), and threat of worsening health for other reasons (2.02 percent vs. 1.92 percent). However, both groups reported rather good health (3.47 percent and 3.46 percent, respectively, on scales 1–4), and moderate trust in government to deal with the COVID-19 crisis (2.91 percent and 3.00 percent on scales 1–4).

### 3.2. Relationships between Resource Threat, Resource Loss, and EWB

To examine the relationships between resource threat, resource loss, and EWB, a structural equation model (SEM) was used, and the results are presented in Table 2. The fit indices of the model were acceptable: χ^2^/df = 1.943; RMSEA = 0.029; CFI = 0.952; IFI = 0.953; standardized root mean square residual SRMR = 0.049.

The study did not find a significant relationship between threat of contracting COVID-19 and EWB. However, a negative significant relationship between worsening health for other reasons and EWB (*β* = −0.204 and *p* < 0.001 for Jews; *β* = −0.172 and *p* = 0.047 for Israeli Palestinians) was found. Thus, hypothesis 1.1 was supported. No significant relationships were found between economic threat and EWB for either Jews or Israeli Palestinians; hypothesis 1.2 was not supported. Social threat was negatively associated with EWB only for Israeli Jews (*β* = −0.124, *p* < 0.001), but not for Israeli Palestinians (*β* = −0.024, *p* = 0.749). Thus, hypothesis 1.3 was supported only for Israeli Jews.

The loss of health resources was negatively associated with EWB for Israeli Jews (*β* = −0.128, *p* < 0.001), but not for Israeli Palestinians (*β* = 0.092, *p* = 0.220). Thus, hypothesis 2.1 was supported only for the majority population. Loss of economic resources was negatively associated with the EWB of both Israeli Jews and Palestinians (*β* = −0.150, *p* < 0.001 for Israeli Jews and *β* = −0.245, *p* = 0.009 for Israeli Palestinians); hypothesis 2.2. was supported. In addition, the loss of social resources in terms of tensions between household members due to the COVID-19 crisis was negatively associated with the EWB of both Israeli Jews and Palestinians (*β* = −0.156, *p* < 0.001 for Israeli Jews and *β* = −0.261, *p* < 0.001). Hypothesis 2.3 was supported.

### 3.3. Comparing Effuects of Resource Threat and Resource Loss on EWB

To test whether the effects of resource threats and losses on EWB varied across the groups, path coefficients were tested for significant differences comparing the models constraining these path coefficients and the unconstrained model (Table 2). The study did not find significant differences in the effects of either health, economic, or social threats on EWB between the groups. Among the effects of losses, the only significant difference was found in effects of health loss (Δχ^2^ = 7.983, *p* = 0.005), where health loss was associated with significantly lower EWB for Israeli Jews (*β* = −0.128, *p* < 0.001) and did not have a significant effect on EWB for Israeli Palestinians (*β* = 0.092, *p* = 0.220). Thus, most effects of threats and losses did not vary across the groups.

To examine whether resource threats or resource losses were associated with a higher decrease in EWB, we compared the path coefficients of threats and losses within each group. A comparison of the effects is presented in Table 3. For both Israeli Jews and Israeli Palestinians, the magnitude of the effect of economic loss was larger than that of economic threat (*β* = −0.150, *p* < 0.001 vs. *β* = 0.008, *p* = 0.844; Δχ^2^ = 6.140, *p* = 0.013 for Israeli Jews; *β* = −0.245, *p* < 0.001 vs. *β* = 0.055, *p* = 0.539; Δχ^2^ = 4.397, *p* = 0.036 for Israeli Palestinians). For Israeli Jews, the magnitude of the effect of health loss (*β* = −0.128, *p* < 0.001) was significantly larger than the magnitude of the effect of health threat for contracting COVID-19 (*β* = −0.012, *p* = 0.747; Δχ^2^ = 11.441, *p* = 0.001). For Israeli Palestinians, the magnitude of the effect of social loss was larger than that of social threat (*β* = −0.261, *p* < 0.001 vs. *β* = −0.024, *p* = 0.749; Δχ^2^ = 4.828, *p* = 0.028). For Israeli Jews and Israeli Palestinians, no differences were found in the effects of health threat from reasons not related to contracting COVID-19 and health loss or in effects of social threat and social loss.

We also compared the effects of threats within each population group. For Israeli Jews, the largest magnitudes were for the effects of health threat for other reasons (*β* = −0.204, *p* < 0.001) and social threat (*β* = −0.124, *p* < 0.001); no difference between them was found (Δχ^2^ = 0.018, *p* = 0.894). However, they were significantly larger than the effects of health threat for contracting COVID-19 (*β* = −0.012, *p* = 0.747) and economic threat (*β* = 0.008, *p* = 0.844). For Israeli Palestinians, no differences between the effects of threats were found.

When comparing the effects of losses, no differences were found in the effects of losses on the EWB of Israeli Jews. However, for Israeli Palestinians, the effects of social loss (*β* = −0.261, *p* < 0.001) and economic loss (*β* = −0.245, *p* = 0.009) were more pronounced than the effect of health loss (*β* = 0.092, *p* = 0.220).

## 4. Discussion

This paper contributes to an emerging body of research on mental health and EWB during times of crisis by explaining which kind of resource loss is salient for predicting EWB decrease and whether the decrease in EWB is mainly associated with actual losses of resources or just from the threat of losses. It transcends the COR theory [15] for times of crisis and expands Bourdieu’s forms of capital [19] by substantiating health as an extremely important additional kind of resource during the current crisis, which began as a health crisis and then spread to encompass the economic and social fields as well. The study revealed that the effects of actual losses are more pronounced in time of crisis than the effects of potential losses (i.e., threats). Another contribution of this study is that it distinguished between the factors predicting EWB for the majority population and an ethnic minority (Israeli Palestinians). An additional contribution of the study is its fusion of the psychological COR theory and the economic law of diminishing marginal utility to explain why loss of resources was more important for predicting EWB for groups that experienced deficient resources before the crisis.

The study found that Israeli Palestinians may be regarded as a disadvantaged minority, because their level of education and rates of employment are lower than those of their Jewish counterparts. In line with previous studies of EWB of minorities [7,9], their EWB was lower than that of the majority population—Israeli Jews. While Israeli Palestinians’ actual economic losses were higher than those of the Jewish population, the social and health losses of both populations were similar. However, Palestinians felt a higher threat from the loss of all kinds of resources than did Israeli Jews. Thus, in terms of the COR model, Israeli Palestinians, an ethnic minority with fewer resources, were more concerned about potential loss of resources due to the pandemic than was the majority population.

As expected, an actual loss of resources was associated with lower EWB for both groups, as was the threat of worsening health for reasons not related to COVID-19. This health threat is probably attributable to the tendency of people to avoid seeking help for their illnesses in public health clinics and hospitals, due to fear of infection. It should be mentioned that during the crisis, telemedicine services in Israel were not expanded to make them available and accessible to the entire. This was due to technical problems related to operating the service and barriers to their use among populations with low digital literacy or no Internet infrastructure [47].

The health threat for contracting COVID-19 and the economic threat had a small and non-significant effect on the EWB of both groups, whereas social threat was associated with lower EWB only for Israeli Jews. One possible explanation might be that the respondents from both groups believed that the low daily number of new cases at that period (May 2020) [39] reflected some control over the situation, a small risk of being infected, and the imminent return of the economic situation to its former state. It is also reasonable to assume that social threat might be less threatening to the EWB of Israeli Palestinians than to that of Israeli Jews. This is because many Palestinians live in multi-generation households, in geographically peripheral and less densely populated cities and villages, and thus the threat was less tangible for them [42].

For both Israeli Jews and Israeli Palestinians, the magnitudes of the effects of resource loss were higher than those of resource threats. The most pronounced effect was for economic resources: economic loss was associated with a higher decrease in EWB for both groups than was economic threat. For Jews, the effect of health loss was higher than the effect of health threat of contracting COVID-19, and for Palestinians, the effect of social loss was higher than the effect of social threat. One possible explanation for the higher effects of losses versus threats is that when actual loss occurs, individuals have to undergo a period of “complicated grief” [6] and mobilize additional resources to reorganize their lives to return to the previous equilibrium. However, threats are less devastating. If faced with the threat of losing economic, social, and health capital [20,23], with all the uncertainty and distress evoked by COVID-19 [31], and hope may prevail, and even if new resources are required, fewer are needed. Threat also may be reinterpreted as a challenge [48] and regarded as a coping strategy: coping requires one to recognize a threat and to decide how to treat the situation [27].

Most effects of threats and losses did not vary across groups. Following the fourth principle of the COR theory [15,49], it may be partly because the disadvantaged minority group such as Israeli Palestinians having outstretched resources, leading them to be defensive and showing some resiliency, such that they could respond, in the survey, to their status closer to the majority privileged Israeli Jews, despite all the odds of disadvantageousness they have to live with. However, in each group, different kinds of losses and threats were salient in predicting EWB. For Israeli Palestinians, economic and social losses were associated with a higher decrease in EWB than health loss. One possible explanation is that coping with the loss of resources requires reserves of resources [15], but their economic resources are relatively limited. The large effect of social loss may be also explained by the extreme importance of social resources for socially segregated ethnic minorities because of the support social-ethnic networks provide [36]. Social-ethnic networks may afford their members many kinds of resources that help them to overcome crisis-related losses. However, Israeli Palestinians were not disadvantaged in health, since they reported the same health status as the Jewish majority. Moreover, among the country’s health professionals they have good representation [50]. Accordingly, the negative effects of actual losses of economic and social resources on EWB were more pronounced than the effects of health loss.

This finding is in line with the COR model [18], according to which people with deficient resources are more vulnerable to additional losses since they do not have enough resources to offset further loss. This is also supported by the law of diminishing marginal utility [35], according to which the loss of a unit of a resource for a person with high amounts of resources decreases the utility (EWB) less than the loss of a unit for a person lacking resources. Thus, for less powerful and disadvantaged groups, depletion of resources is more important for predicting their EWB.

For Israeli Jews, no differences were found in the effects of losses on EWB; however, differences appeared in the effects of threats. Effects of the threat of worsening health for reasons not related to contracting COVID-19 and the social threat had the largest magnitudes, compared to the effects of the threat of contracting COVID-19 and economic threat. Thus, for the majority, concerns unrelated directly to contracting COVID-19 were better predictors of EWB than those who were worried about being infected with the coronavirus. In other words, it is possible to deny the seriousness of COVID-19, but not the potential losses it can cause. For Israeli Palestinians, no differences were found in the effects of threats on EWB, all of which were relatively small.

Limitations: It is widely reported that among the wider impacts of COVID-19 on populations, the exacerbation of already existing socio-economic disadvantageousness has been a global pattern [51,52]. There may be some confounding impact of exacerbation of pre-existing disadvantageousness of Israeli Palestinians on EWB with the actual impact of COVID-19 on EWB through resource losses or threats among them. For example, the survey did not contain information about the absolute income of respondents before, and during the crisis, it only asked them about the change in income during the crisis. It also did not include data as to a loss of the inter-family social interactions and data allowing for a distinction between other minority groups within the Jewish population. The study could not examine the long-term consequences of the pandemic, for which longitudinal studies are needed. Furthermore, the data were gathered on 11–14 May 2020 at the end of the first wave. Since then, Israel has suffered a much more severe wave of morbidity and mortality, which may influence the magnitude of the effect of loss and threat on EWB in both groups. In addition, the study was quantitative and further qualitative research may contribute to a deeper understanding of differences between the impact of threats and losses on the EWB of minorities.

## 5. Conclusions

During the pandemic, the actual economic losses of Israeli Palestinians, an ethnic minority, were higher than those of the Jewish majority population. The social and health losses of both populations were similar. However, Israeli Palestinians, an ethnic minority with fewer resources, were more concerned about the potential loss of resources than the majority population. The study revealed that the effects of actual losses on EWB are more pronounced in time of crisis than the effects of potential losses (i.e., threats). The largest effects were for economic resources. For ethnic minorities, who are less powerful and more disadvantaged than ethnic majorities, the depletion of already deficient resources during time of crisis is more important for predicting their EWB than for the majority population.

The findings of this study are relevant not only for the COVID-19 pandemic period, but also for explaining factors predicting EWB in times of any crisis (economic, social, and military) that may occur. The fact that Israeli Jews and Palestinians, groups in protracted conflict, reported similar levels of trust in how the government is dealing with the COVID-19 crisis indicates that shared struggles, threats, and losses implied by the crisis may increase the confidence and trust of minorities in a government. The findings of the study highlight the importance of developing tailored economic and social support interventions for minorities in times of resource losses and threats, which may help decision-makers to develop a more customized policy during crisis.

## Figures and Tables

**Table 1 ijerph-18-12590-t001:** Descriptive statistics for the sample.

Variable	Israeli Jews	Israeli Palestinians
Sex (% male)	46.9	48.9
Age (%):		
21–44	42.5	54.0
45–64	30.4	31.6
65+	27.1	14.4
Married (%)	71.1	71.8
Education (% academic education)	33.6	12.1
Employment status (%):		
Salaried worker	43.4	35.7
Self-employed	12.0	7.0
Furloughed	8.6	10.5
Not employed	35.9	46.8
Health status, scales 1–4, mean (SD)	3.47 (0.77)	3.46 (0.89)
Trust in government to deal with the COVID-19 crisis, scale 1–4, mean (SD)	2.91 (1.05)	3.00 (1.07)
EWB index, based on 3 items, scales 1–4, higher values indicate better EWB, mean (SD)	3.36 (0.77)	3.23 (0.95)
Feel pressure and anxiety (%)		
not at all	48.4	51.1
not so much	18.9	14.4
to some extent	25.4	16.7
to a great extent	7.3	17.8
Feel depressed (%)		
not at all	71.0	65.9
not so much	14.3	10.4
to some extent	11.1	13.3
to a great extent	3.6	10.4
Feel lonely (%)		
not at all	68.5	70.5
not so much	13.4	8.1
to some extent	13.3	10.4
to a great extent	4.7	11.0
Resource threats due to the COVID-19, scales 1–4:		
Health threat of contracting the COVID-19, mean (SD)	2.42 (1.04)	2.54 (1.26)
Health threat of worsening health for other reasons, mean (SD)	1.92 (1.03)	2.02 (1.17)
Economic threat of difficulties in meeting expenses, mean (SD)	2.20 (1.16)	2.50 (1.26)
Social threat of worsening family relationships, mean (SD)	1.30 (0.61)	1.47 (0.96)
Resource losses due to the COVID-19, scales 1–5:		
Health loss—health has worsened, mean (SD)	2.97 (0.36)	2.98 (0.63)
Economic loss—economic situation of household has worsened, mean (SD)	3.45 (0.64)	3.63 (0.72)
Social loss—tensions between household members, mean (SD)	1.62 (0.85)	1.65 (1.09)

**Table 2 ijerph-18-12590-t002:** Results of multigroup SEM ^a^—standardized path coefficients of the model predicting emotional well-being (EWB).

Variables	Standardized Effects		Effect Comparison: Israeli Jews vs. Israeli Palestinians
	Israeli Jews	Israeli Palestinians	
Resource threats due to the COVID-19 crisis:			
Health threat of contracting COVID-19	−0.012	−0.097	Δχ^2^ = 0.634, *p* = 0.426
Health threat of worsening health for other reasons	−0.204 ***	−0.172 *	Δχ^2^ = 0.001, *p* = 0.979
Economic threat of difficulties in meeting expenses	0.008	−0.055	Δχ^2^ = 0.393, *p* = 0.531
Social threat of worsening family relationships	−0.124 ***	−0.024	Δχ^2^ = 1.654, *p* = 0.198
Resource loss due to the COVID-19 crisis:			
Health loss—health has worsened	−0.128 ***	0.092	Δχ^2^ = 7.983, *p* = 0.005
Economic loss—economic situation of household has worsened	−0.150 ***	−0.245 **	Δχ^2^ = 1.307, *p* = 0.253
Social loss—tensions between household members	−0.156 ***	−0.261 ***	Δχ^2^ = 1.620, *p* = 0.203
Control variables:			
Sex (1 = male)	0.067 *	−0.025	
Age (reference group = 21–44):			
Age 45–64	0.039	0.030	
Age 65+	0.083 *	0.133	
Education	−0.008	0.015	
Married	0.040	−0.063	
Employment status (reference group = non-employed):			
Salaried worker	0.062	0.084	
Self-employed	0.041	−0.044	
Furloughed	0.039	−0.011	
Health status	0.226 ***	0.368 ***	
Trust in the government to deal with the COVID-19 crisis	0.049	0.136 *	

^a^ χ^2^/df = 1.943; RMSEA = 0.029; CFI = 0.952; incremental fit index IFI = 0.953; SRMR = 0.048. *** Significance < 0.001; ** significance < 0.010; * significance < 0.05.

**Table 3 ijerph-18-12590-t003:** Comparison of effect magnitudes on EWB.

Compared Effects	Israeli Jews	Israeli Palestinians
Threats vs. losses:		
Health threat for contracting COVID-19 vs. health loss	Δχ^2^ = 11.441, *p* = 0.001 *β* = −0.012 vs. *β* = −0.128 ***	Δχ^2^ = 2.134, *p* = 0.144 *β* = −0.097 vs. *β* = 0.092
Health threat for other reasons vs. health loss	Δχ^2^ = 2.457, *p* = 0.117 *β* = −0.204 *** vs. *β* = −0.128 ***	Δχ^2^ = 3.658, *p* = 0.056 *β* = −0.172 * vs. *β* = 0.092
Economic threat vs. economic loss	Δχ^2^ = 6.140, *p* = 0.013 *β* = 0.008 vs. *β* = −0.150 ***	Δχ^2^ = 4.397, *p* = 0.036 *β* = 0.055 vs. *β* = −0.245 ***
Social threat vs. social loss	Δχ^2^ = 0.005, *p* = 0.943 *β* = −0.124 *** vs. *β* = −0.156 ***	Δχ^2^ = 4.828, *p* = 0.028 *β* = −0.024 vs. *β* = −0.261 ***
Threats vs. threats:		
Health threat for contracting COVID-19 vs. health threat for other reasons	Δχ^2^ = 9.947, *p* = 0.002 *β* = −0.012 vs. *β* = −0.204 ***	Δχ^2^ = 0.242, *p* = 0.623 *β* = −0.012 vs. *β* = −0.172 *
Health threat for contracting COVID-19 vs. economic threat	Δχ^2^ = 0.118, *p* = 0.731 *β* = −0.012 vs. *β* = 0.008	Δχ^2^ = 0.087, *p* = 0.769 *β* = −0.097 vs. *β* = −0.055
Health threat for contracting COVID-19 vs. social threat	Δχ^2^ = 7.164, *p* = 0.007 *β* = −0.012 vs. *β* = −0.124 ***	Δχ^2^ = 0.213, *p* = 0.644 *β* = −0.097 vs. *β* = −0.024
Health threat for other reasons vs. economic threat	Δχ^2^ = 13.805, *p* = 0.000 *β* = −0.204 *** vs. *β* = 0.008	Δχ^2^ = 0.730, *p* = 0.393 *β* = −0.172 * vs. *β* = −0.055
Health threat for other reasons vs. social threat	Δχ^2^ = 0.018, *p* = 0.894 *β* = −0.204 *** vs. *β* = −0.124 ***	Δχ^2^ = 0.965, *p* = 0.326 *β* = −0.172 * vs. *β* = −0.024
Economic threat vs. social threat	Δχ^2^ = 4.828, *p* = 0.028 *β* = 0.008 vs. *β* = −0.124 ***	Δχ^2^ = 0.029, *p* = 0.865 *β* = −0.055 vs. *β* = −0.024
Losses vs. losses:		
Health loss vs. economic loss	Δχ^2^ = 0.861, *p* = 0.353 *β* = −0.128 *** vs. *β* = −0.150 ***	Δχ^2^ = 5.755, *p* = 0.016 *β* = −0.092 vs. *β* = −0.245 ***
Health loss vs. social loss	Δχ^2^ = 2.794, *p* = 0.095 *β* = −0.128 *** vs. *β* = −0.156 ***	Δχ^2^ = 6.253, *p* = 0.012 *β* = −0.092 vs. *β* = −0.261 ***
Economic loss vs. social loss	Δχ^2^ = 0.431, *p* = 0.511 *β* = −0.150 *** vs. *β* = −0.156 ***	Δχ^2^ = 0.470, *p* = 0.493 *β* = −0245 *** vs. *β* = −0.261 ***

*** Significance < 0.001; * Significance < 0.05.

## Data Availability

The raw data of the representative 2020 Corona Survey administered in Israel is available at: https://doi.org/10.6084/m9.figshare.15123849.v1 (accessed on 26 November 2021).

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
