# Peer review of "Resource Threat versus Resource Loss and Emotional Well-Being of Ethnic Minorities during the COVID-19 Pandemic"

_ijerph, 2021, doi:10.3390/ijerph182312590_

Round 1

Reviewer 1 Report

General Comments:

This is an important piece of work on a number of novel grounds, namely studying emotional well being (EBW) beyond mental health issues, grounded on the conservation of resources theory, less studied minority Israeli Palestinians compared to the dominant group of Israeli Jews, and various risk factors associated with conversation of resources during the COVID-19 pandemic crises.

Specific comments:

Line 28: Better to mention, “The ongoing COVID-19 pandemic is an example….”

Line 78: Better to replace “…since they have fewer resources…” with “…. since they are left with…”

Lines 227-232: Attributing ‘social loss’ to intra-family tensions may be somewhat weak, as much of the social loss may happen to due loss across the inter-family social interactions.

Lines 293-300: This study did not find significant differences in most effects of threats and losses between Israeli Jews and Israeli Palestinians. Following the fourth principle of Conservation of Resources theory (Hobfoll et al., 2018), it may be partly because, the disadvantaged minority group like Israeli Palestinians having outstretched resources leading them to be defensive showing some resiliency such that they could respond, in the survey, about their status closer to the majority privileged Israeli Jews, despite all the odds of disadvantageousness they have to live with.

Lines 267-271: The results of the observed variables using the SEM analyses are presented well. In addition, the factor analysis using SEM could generate derived latent variables to summarize the effects at higher level of interpretation.

Line 379: typo error, uncertainty?

Lines 413-422: It is widely reported that among the wider impacts of COVID-19 on populations, exacerbation of already existing socio-economic disadvantageousness has been a global pattern. There may be some confounding impact of exacerbation of pre-existing disadvantageousness of Israeli Palestinians on EWB with the actual impact of COVID-19 on EWB through resource losses or threats among them.

Table 1: 1) Simply considering biological sex, male and female, may not represent wholly the gender phenomena that exist between two ethnic groups that may have differences in gender values. Gender analysis could be described in the discussion. 2) Education levels require bit clarification of what academic vs. non-academic are and at what level. 3) Higher percentage of Israeli Palestinians feeling pressure and anxiety to a great extent compared to Israeli Jews, and reverse patterns of percentages in feeling depressed and feeling lonely between the two groups, may indicate greater resiliency among the former than the latter in coping with forces against the odds for EWB during COVID-19 pandemic. Larger and multigenerational households in the former may provide some cushion (means) for gaining this resiliency.

Table 1 & 2: Please show the notations, *, **, *** what they indicate in the foot of the table. Define all abbreviations and mathematical symbols.

Author Response

Nov. 27, 2021

Answer to reviewer 1

Reviewer 1

This is an important piece of work on a number of novel grounds, namely studying emotional well being (EBW) beyond mental health issues, grounded on the conservation of resources theory, less studied minority Israeli Palestinians compared to the dominant group of Israeli Jews, and various risk factors associated with conversation of resources during the COVID-19 pandemic crises.

Thank you for this comment

Specific comments:

Line 28: Better to mention, “The ongoing COVID-19 pandemic is an example….”

We have changed the sentence accordingly: " The ongoing COVID-19 pandemic is an example of such a crisis… ".

Line 78: Better to replace “…since they have fewer resources…” with “…. since they are left with…”

 We have changed the sentence accordingly: " since they are left with fewer resources to cope with new challenges [22]."

Lines 227-232: Attributing ‘social loss’ to intra-family tensions may be somewhat weak, as much of the social loss may happen to due loss across the inter-family social interactions.

We agree with this comment. Unfortunately, 2020 Corona Survey we used did not contain information about the loss across the inter-family social interactions. This survey was conducted the Israel Central Bureau of Statistics, so that we could use only questions it contained. We added this issue to study limitations.

Lines 293-300: This study did not find significant differences in most effects of threats and losses between Israeli Jews and Israeli Palestinians. Following the fourth principle of Conservation of Resources theory (Hobfoll et al., 2018), it may be partly because, the disadvantaged minority group like Israeli Palestinians having outstretched resources leading them to be defensive showing some resiliency such that they could respond, in the survey, about their status closer to the majority privileged Israeli Jews, despite all the odds of disadvantageousness they have to live with.

Thank you for this comment. We have added this explanation to the beginning of paragraph six in the discussion, (just after the sentence "Most effects of threats and losses did not vary across groups").

Lines 267-271: The results of the observed variables using the SEM analyses are presented well. In addition, the factor analysis using SEM could generate derived latent variables to summarize the effects at higher level of interpretation.

The factor structure and measurement invariance of the latent variable EWB were justified by means of Confirmatory Factor Analysis (CFA). The fit indices of the configural, metric, and scalar models were found to be acceptable (χ2/df < 1.575, RMSEA < 0.023, CFI > 0.998, NFI > 0.992). All item weights were higher than 0.668. The differences in Chi-square values between the models were found to be non-significant (Δχ2 [2] = 0.887, p = 0.642 between the baseline and metric models, and Δχ2 [3] = 6.983, p = 0.072 between the metric and scalar models). According to Davidov et al. [46], three invariance levels – configural, metric and scalar – are evidence of measurement invariance. Thus, the constructs of EWB can be legitimately compared across the groups.

We also generated derived latent variable EWB and presented its value in Table 1.

These explanations are presented in the beginning of Results section.

Line 379: typo error, uncertainty?

This typo has been corrected.

Lines 413-422: It is widely reported that among the wider impacts of COVID-19 on populations, exacerbation of already existing socio-economic disadvantageousness has been a global pattern. There may be some confounding impact of exacerbation of pre-existing disadvantageousness of Israeli Palestinians on EWB with the actual impact of COVID-19 on EWB through resource losses or threats among them.

It is a very valid point, thank you. We have changed the text of the 'limitation' paragraph as follows: "It is widely reported that among the wider impacts of COVID-19 on populations, exacerbation of already existing socio-economic disadvantageousness has been a global pattern [51,52]. There may be some confounding impact of exacerbation of pre-existing disadvantageousness of Israeli Palestinians on EWB with the actual impact of COVID-19 on EWB through resource losses or threats among them. For example, the…". Also, we added two additional references to support the claim [51,52].   

Table 1: 1) Simply considering biological sex, male and female, may not represent wholly the gender phenomena that exist between two ethnic groups that may have differences in gender values. Gender analysis could be described in the discussion. 2) Education levels require bit clarification of what academic vs. non-academic are and at what level. 3) Higher percentage of Israeli Palestinians feeling pressure and anxiety to a great extent compared to Israeli Jews, and reverse patterns of percentages in feeling depressed and feeling lonely between the two groups, may indicate greater resiliency among the former than the latter in coping with forces against the odds for EWB during COVID-19 pandemic. Larger and multigenerational households in the former may provide some cushion (means) for gaining this resiliency.

  1. You are right. The survey did not contain an information about gender, only about sex. We changed Gender to Sex through the paper.
  2. We added a clarification about education levels in the Method section.
  3. Thank you very much for putting attention to this controversial data! It was a kind of typo mistake, we just confused between the columns. Israeli Palestinian reported feeling more pressure and anxiety, and also feeling more depressed and more lonely than the Israeli Jews. This is consistent with the value of computed EWB index, which was better for the Jews than for Palestinians. We fixed this mistake and changed the data in the Table 1 accordingly.

Table 1 & 2: Please show the notations, *, **, *** what they indicate in the foot of the table. Define all abbreviations and mathematical symbols.

We added this information in the foot pf the tables 2 and 3.

Reviewer 2 Report

Review Report

            The manuscript “Resource Threat versus Resource Loss and Emotional Well-Being of Ethnic Minorities during the COVID-19 Pandemic” provides relevant information for the health-care professionals regarding the mental health and the well-being of the population during the COVID-19 pandemic. This study is well documented by the multiple articles cited and has a rigorous methodology.

My observations are:

  1. Please make sure that the word “Black” (page 1, line 44) is the correct and ethical form to describe someone’s ethnical provenience. I suggest using instead, the term “Afro-American”.
  2. In page 2, line 54, I suggest the word “vs” to be expanded, as this is a scientific paper. The same for page 2, line 62 etc.
  3. In page 3, lines 133-134 you didn’t provide references for all the studies mentioned “other studies”.
  4. Please ensure that the abbreviation “U.S.” in page 1, line 44 was expanded above.
  5. For paragraph between, lines 49 and 56, page 2, please provide references for the “numerous” and “several” studies cited in the manuscript.
  6. For the paragraph between lines 124 and 130, page 3, is redundant to use the same reference twice “[6]”. Please make sure that all the information in this paragraph cand be found in the reference [6]. If your citation is correct, please use only one citation after the paragraph.
  7. In page 4, for the paragraph between 167-175 “the virus… at once” you didn’t provide any reference.

In conclusion the manuscript requires minor revision in terms of bibliography.

Author Response

Nov. 26, 2021

Answer to reviewer 2

The manuscript “Resource Threat versus Resource Loss and Emotional Well-Being of Ethnic Minorities during the COVID-19 Pandemic” provides relevant information for the health-care professionals regarding the mental health and the well-being of the population during the COVID-19 pandemic. This study is well documented by the multiple articles cited and has a rigorous methodology.

My observations are:

  1. Please make sure that the word “Black” (page 1, line 44) is the correct and ethical form to describe someone’s ethnical provenience. I suggest using instead, the term “Afro-American”.

We have changed the words accordingly. 

  1. In page 2, line 54, I suggest the word “vs” to be expanded, as this is a scientific paper. The same for page 2, line 62 etc.

We have expanded the word in the places mentioned and in the discussion chapter.

  1. In page 3, lines 133-134 you didn’t provide references for all the studies mentioned “other studies”.

There was an error typing. We refer to a paper published by the American Psychological Association whose details appear in the reference list under the heading 'Grief and COVID-19: Mourning our bygone lives'. We changed the sentence as follows: "On the contrary, Weir [32] claims that whether the loss is actual or anticipated, the reactions are genuine and valid. "

  1. Please ensure that the abbreviation “U.S.” in page 1, line 44 was expanded above.

We have expanded the abbreviation U.S. with the full name, United States.

  1. For paragraph between, lines 49 and 56, page 2, please provide references for the “numerous” and “several” studies cited in the manuscript.

We have added the relevant reference to the “numerous” [2,3,5] and to “several” studies [16,17].

  1. For the paragraph between lines 124 and 130, page 3, is redundant to use the same reference twice “[6]”. Please make sure that all the information in this paragraph cand be found in the reference [6]. If your citation is correct, please use only one citation after the paragraph.

The reference is relevant, and we use only one citation after the paragraph.

  1. In page 4, for the paragraph between 167-175 “the virus… at once” you didn’t provide any reference.

Indeed, we have added a reference: Last, M. The first wave of COVID-19 in Israel – Initial analysis of publicly available data. PloS one 2020, 15(10), e0240393. https://doi.org/10.1371/journal.pone.0240393 [39]

In conclusion the manuscript requires minor revision in terms of bibliography.
